# *Triatoma yelapensis* sp. nov. (Hemiptera: Reduviidae) from Mexico, with a Key of *Triatoma* Species Recorded in Mexico

**DOI:** 10.3390/insects14040331

**Published:** 2023-03-29

**Authors:** Juan Téllez-Rendón, Lyda Esteban, Laura Rengifo-Correa, Héctor Díaz-Albiter, Herón Huerta, Carolina Dale

**Affiliations:** 1Institute of Epidemiological Diagnosis and Reference (InDRE), Mexico City 01480, Mexico; 2Tropical Diseases Research Center (CINTROP), Universidad Industrial de Santander, Piedecuesta 681012, Colombia; 3Department of Health, The School of the Southern Border (ECOSUR), Villahermosa 86280, Mexico; 4Laboratório de Biodiversidade Entomológica, Instituto Oswaldo Cruz, Fundação Oswaldo Cruz, Rio de Janeiro 4365, Brazil

**Keywords:** Triatominae, *Triatoma recurva*, geometric morphometry, vector of Chagas disease

## Abstract

**Simple Summary:**

Triatominae (Hemiptera, Reduviidae) bugs are considered species of medical relevance because they are vectors of Chagas disease. In Mexico, most species of the *Triatoma* genus are distributed in the lowlands of the west of Mexico, including the Mexican Pacific coast. Here, we describe *Triatoma yelapensis* sp. nov. from the Pacific coast of Jalisco (Mexico). We provide statistical support for the morphological distinctiveness of the new species and an updated key of the genus *Triatoma* for species recorded in Mexico.

**Abstract:**

Thirty-four species of Triatominae (Hemiptera, Reduviidae) are recorded in Mexico, *Triatoma* Laporte, 1832 the most speciose genus in this country. Here, we describe *Triatoma yelapensis* sp. nov. from the Pacific coast of Jalisco (Mexico). The most similar species to *T. yelapensis* sp. nov. is *T. recurva* (Stål, 1868), but they differ in head longitude, the proportion of labial segments, coloration pattern of corium and connexivum, spiracles location, and male genitalia. To provide statistical support for the morphological distinctiveness of the new species, we performed a geometric morphometric analysis of *T. yelapensis* sp. nov., *T. dimidiata s.s.* (Latreille, 1811), *T. gerstaeckeri* (Stål, 1859), and *T. recurva* (Stål, 1868), considering head morphology. We also provide an updated key of the genus *Triatoma* for species recorded in Mexico.

## 1. Introduction

Around 7 million people worldwide are currently infected with the flagellated parasite *Trypanosoma cruzi* (Chagas 1909)—the etiological agent of Chagas disease—with 30,000 new cases annually [1,2]. In Mexico, it is estimated that 1.1 million people are infected [3]. Transmission of *T. cruzi* occurs mainly by contact with infected feces of Triatominae (Hemiptera, Reduviidae) bugs [3]. This subfamily includes approximately 153 extant species and 3 fossils, distributed across 5 tribes and 18 genera [4,5,6,7,8,9,10,11,12,13,14,15]. In Mexico, seven genera (*Belminus* Stål, 1859; *Dipetalogaster* Usinger, 1939; *Eratyrus* Stål 1859; *Panstrongylus* Berg, 1879; *Paratriatoma* Barber, 1938; *Triatoma* Laporte, 1832; and *Rhodnius* Stål, 1859) and 34 species are recorded [16,17]. In this country, the transmission of Chagas disease is mainly attributed to some members of the *Triatoma phyllosoma* species group, which currently includes 17 species [18,19].

Most species of the *T. phyllosoma* species group are distributed in the lowlands of the west of Mexico, including the Mexican Pacific coast. In this area, Jalisco stands out among other states because of its large richness of species of the *T. phyllosoma* species group, high rates of natural infection of *T. cruzi* in these triatomines and prevalence of Chagas disease in humans [20,21]. Seven species of the *T. phyllosoma* species group are recorded in Jalisco: *T. brailovskyi* Martínez, Carcavallo and Peláez, 1984, *T. dimidiata*, *T. mazzottii* Usinger 1941, *T. longipennis* Usinger 1939, *T. pallidipennis* (Stål, 1872), *T. phyllosoma* (Burmeister, 1835), and *T. picturata* Usinger, 1939 [18,19,20,22,23,24,25,26]. However, records of *T. mazzottii* in Jalisco are controversial [19,22].

In 2006, the Institute of Epidemiological Diagnosis and Reference (InDRE) processed several specimens of triatomines from Jalisco, whose morphological characters were different from any other known species of *Triatoma*. Considering these specimens, here we describe a new species of the genus *Triatoma* Laporte, 1832. This new species seems closely related to the *T. phyllosoma* species group, particularly resembling *T. recurva* (Stål, 1868). We provide statistical support for the morphological distinctiveness of the new species using geometric morphometry. Finally, we provide an updated key of species of the genus *Triatoma* recorded in Mexico.

## 2. Materials and Methods

The Secretary of Health of the State of Jalisco sent eight *Triatoma* sp. specimens to the InDRE Entomology Laboratory. These specimens were collected manually, from 2006 to 2017, after performing entomological surveillance in a human settlement called “Selva El Tuito”. Specimens were collected from Yelapa (20.4875 N, −104.5536 W) and Pizota (20.4900 N, −105.4892 W) localities, in Cabo Corrientes, Jalisco, Mexico. These sampled places are located 25 km from Puerto Vallarta and are isolated by a mountain range of the largest continental portion of Jalisco (Figure 1). These areas have a mean annual temperature of 25.6 °C and rainfall of 878.3 mm^3^ [27]. Voucher specimens are preserved in the Colección de Artrópodos de Importancia Médica (CAIM) of the Instituto de Diagnóstico y Referencia Epidemiológicos (InDRE).

We use the following terminology: general morphology of Lent and Wygodzinsky [7] and Weirauch [28], antennal sensilla of May-Concha et al. [29], and genitalia of Lent and Wygodzinsky [7] and Zhao et al. [13]. Habitus images were obtained using a Nikon D100 camera, with a 55 mm macro lens. Detailed observations and measurements of the head (1.6×), pronotum (1×), wings (1×), legs (1×), and male genitalia (6.3×) were made using a Zeiss Discovery V8 stereoscope and an Axion Vision V 4.8.2.0 camera. For male genitalia drawings, we used a Zeiss Stemi V6 stereoscope with a light camera at 5× magnification. Drawings were edited using Adobe Illustrator v.24.3 and Photoshop^®^ software.

To provide statistical support for the morphological distinctiveness of the new species, we performed a geometric morphometric analysis of *T. yelapensis* sp. nov., *T. dimidiata s.s.*, *T. gerstaeckeri* (Stål, 1859), and *T. recurva*, considering head morphology [30]. The last three species were chosen for this analysis because they morphologically resemble *T. yelapensis* sp. nov. more than other species of the *T. phyllosoma* complex. Additionally *T. dimidiata s.s.*, *T. gerstaeckeri*, and *T. recurva* come, respectively, from the three main clades of the *T. phyllosoma* species group [20]. We chose 12 landmarks of the dorsal view of the head (Figure 2A) for the following specimens: *T. yelapensis* sp. nov. (3 males, 5 females), *T. recurva* (3 males from Chihuahua; 6 males, 8 females from Sinaloa), *T. dimidiata s.s.* (10 males, 16 females from Chiapas; 5 males from Yucatán), and *T. gerstaeckeri* (2 males, 5 females from Tamaulipas; 1 male from San Luis Potosi; 1 male from Puebla; 1 male, 2 females from Hidalgo; 2 males from Coahuila). Landmarks were located in homologue structures of previously referenced landmarks for triatomines [4,13], which are easily detectable among specimens. We selected 12 landmarks: 4 of type I and 8 of type II. Landmarks were captured using the XYOM-CLIC V 99 software [31]. Males and females were analyzed together instead of each genus on its own because of the low number of specimens available for some examined species. Anatomical landmarks were processed using a generalized procrustes analysis (GPA). This analysis orthogonally aligns and projects residual Procrustes onto a flat plane tangent to the consensus of the aligned objects [31]. Residual Procrustes of the 69 studied specimens were retrieved from the superimposition of Procrustes and used as input for the analysis of principal components (ACP). The first six principal components were used as final conformation variables for canonical variate (CVA) analysis. ANOVA statistics were used to compare the variance between group means for centroid size (CS). Analyses were performed using the XYOM V2 online software available at https://xyom.io (accessed on 14 May 2021).

## 3. Results

### 3.1. Taxonomy


Familia Reduviidae Latreille, 1807Subfamilia Triatominae Jeannel, 1919Genero *Triatoma* Laporte, 1832*Triatoma yelapensis* sp. nov.(Figure 2, Figure 3, Figure 4, Figure 5 and Figure 6)


Type material. Holotype: one male, MEXICO, Jalisco, Cabo Corrientes, Yelapa. 20.4875 N, −104.5536 W. 1.3 masl. 29 May 2006. Peridomicile. CAIMTriTp-0006, deposited in Colección de Artrópodos de Importancia Médica, of the Instituto de Diagnóstico y Referencia Epidemiológicos (InDRE). Paratypes: one female, same data as holotype except collection code: CAIMTriTp-0007. CAIM. One female, same data as holotype except 23 April 2007, CAIMTriTp-0008. One female, same data as holotype except 6 May 2008, CAIMTriTp-0005. Two males, same data as holotype except 16 May 2013, CAIMTriTp-0003, CAIM. CAIMTriTp-0002. One female, same data as holotype except 2 February 2015, CAIMTriTp-0009. One female, MEXICO, Jalisco, Cabo Corrientes, Pizota. 20.4900 N, −105.4892. 1.3 masl. 26 May 2011. Peridomicile, CAIMTriTp-0004.

### 3.2. Description

#### 3.2.1. Coloration

General coloration dark brown for the holotype, to black. Head uniformly black, except for a reddish-brown narrow area from maxillary plate to clypeus (Figure 2B); reddish-brown narrow area is more evident in some specimens than in others. Antenna color: pedicellus dark brown in the apex, slightly lighter basally; basiflagellomere uniformly dark brown; distiflagellomere light brown in the apex and darker basally (Figure 2C). Neck dark brown to black, with a yellowish spot laterally. Labium dark reddish brown; connections between labial segments light yellow (Figure 2D). Pronotum uniformly dark brown (Figure 3A). Scutellum dark brown; central depression dark reddish brown; posterior process dark brown. Hemelytra with corium yellow; large dark central spot no surpassing R and R + M veins and extending approximately to the basal area of corium (Figure 3A); apex dark brown; costal margin basally dark brown; clavus basally black and apically smoked black; membrane smoky brown, lighter than dark brown portion of corium (Figure 3A); a cloudy spot dark brown between cubitus and postocubitus veins. Legs uniformly dark. Connexivum dark brown; subtriangular yellow-orange spot on dorso-ventral external margin, wide in posterior half, reaching the connexival suture (Figure 3 and Figure 4). Sternites uniformly dark brown (Figure 4). Spiracles yellowish, surrounded by a yellow spot on the tegument (Figure 4).

#### 3.2.2. Morphological Features

Female total length 30–31 mm and male, 28–29 mm. Head longer than wide (1:0.38–0.40). Head length longer than the length of the pronotum (1:0.7–0.8). Ocellar lens 0.20–0.23 mm; width of eye shorter than synthlipsis (1:1.58–1.75). Ratio of labial segments 1:1.54–1.77:0.41–0.45. Pronotum glabrous. Mesosternum with a transverse T-shaped knob on median portion. Spiracles adjacent to connexival suture (Figure 4). Fossula spongiosa present on fore tibia of males, absent in females. Pygophore globular, rugose transversally, marginate distally. Discoid-shaped and denticulate in anterior half, medium size (0.45 mm) (Table 1).

Head. Cylindrical, narrow, rugose transversally on dorsum and irregular laterally, slightly rugose below, 2.5 times as long as wide across eyes (1:0.40); anteocular region 3 times as long as postocular (1:0.3). Eyes in lateral view reaching ventral but not dorsal margin of head. Ratio between width of eye and synthlipsis 1:1.59. Ocellar lens small (0.23 mm) located at U-shaped tubercle with subparallel sides. Antenniferous tubercles situated in the middle of the anteocular region, not surpassing the mandibular plate. First antennal segment not surpassing the apex of clypeus. Labium slender; first visible segment extending but not surpassing level of the base of antenniferous tubercle; second visible segment extending to the level of the yellow spot on neck, attaining level of base of head. Ratio of labial segments 1:1.73:0.45. Clypeus long and narrow apically. Maxillary plate not attaining apex of clypeus, slightly sharped apically (Figure 2B). Anterolateral projections large, subconical, rounded. Head longer than pronotum (1:0.77).

Pronotum. Constricted at the level of transversal sulcus. Disc of anterior lobe granulose with 1 + 1 discal tubercles and 1 + 1 lateral tubercles, smaller than half of length of discal tubercles, but still perceptible. Posterior lobe heavily wrinkled; submedian carinae on posterior half of lobe; humeral angles rounded. Scutellum triangular heavily rugose with central portion depressed; apical process elongate sub-cylindrical and apically rounded. Mesosternum with a knob transverse T-shaped on median portion (Figure 4C). Hemelytra surpassing urotergite VI, not attaining VII margin in either sex, leaving entire connexivum and lateral portions of urotergites exposed (Figure 3A). Spiracles on abdominal sternites, adjacent, but not touching connexival suture (Figure 4B).

Legs. Slender and sub-cylindrical; fore femora with a row of five denticles in small elevations on anterolateral portion; fore and mid femora with one pair of small denticles subapically under surface. Males with small fossula spongiosa on fore tibia; legs of females without fossula spongiosa. Abdomen one-third as wide as total length of body, convex slightly flattened ventrally in the middle (Figure 3A).

Vestiture. Setae decumbent, sparse, inconspicuous (0.1–0.2 mm). Head with setae very sparse and decumbent. Pedicellus and basiflagellomere of antenna with short and decumbent setae; distiflagellomere with bristles erected and numerous sub-erect sensilla trichoidea. Second labial segment with few inconspicuous setae; apex of third labial segment with conspicuous setae. Pronotum and scutellum glabrous. Mesosternum and metasternum with scattered and very sparse short setae, decumbent. Hemelytra glabrous. Lateral margin of connexivum with dark short and thick setae; female with abundant setae on seventh segment. Pilosity of venter short and decumbent.

Male genitalia (Figure 5 and Figure 6). Pygophore globose, rugose transversally in the posterior third on venter, marginate distally; general color piceo; in lateral view longer than width (1:0.7); transverse bridge of pygophore sclerotized and large. Median process of pygophore triangular, short, not depressed at baseline, with long setae. Parameres elongated, cylindrical, subapically curved; rugose in the externo-apical portion; setae very scarce on both surfaces ventral and dorsal. Articulatory apparatus long. Basal plate extension rectangular, shorter than basal plate (1:0.83), strongly curvated in lateral view; arms elongated, divergent but united by a somewhat narrow and slightly elongated basal bridge. Gonopore process cylindrical, hollow, elongated, occupying the entire medial extension of basal plate, lateral margin sclerotized. Phallosoma support with two elongated lateral arms, joined at basal portion and separated at apex. Medial basal sclerite of phallosoma, large, rounded at apex, strongly sclerotized. Dorsal phallotecal sclerite ovoid, laminar plate with parallel sides, basally opening, less sclerotized on the top. Endosomal process medium to small size, discoid-shaped, sclerotized, and denticulate in the anterior half.

### 3.3. Etymology

The specific epithet derives from the type locality, Yelapa, and the Latin adjectival suffix “-ensis”, meaning “originating in” or “pertaining to”.

### 3.4. Host–Parasite Data

Only one specimen of *Triatoma yelapensis* sp. nov. was examined for *Trypanosoma cruzi* infection using the PCR technique and the following primers: Fw: GCAGTCGGCKGATCGTTTTCG; Rv: TTCAGRGTTGTTTGGTGTCCAGTG. The evaluated sample was positive for *T. cruzi* infection (Figure 7).

### 3.5. Species Group Assignment

The combination of characters suggested by Rengifo-Correa et al. [20] to diagnose the *Triatoma phyllosoma* species group was observed in *T. yelapensis* sp. nov. The following character combination is shared between *T. yelapensis* sp. nov. and the *Triatoma phyllosoma* species group members: head slightly longer than pronotum; location of annteniferous tubercles in the middle of the anteocular region, not surpassing the mandibular plate; genae tapering apically, not surpassing apex of clypeus; first and third rostral segments much shorter than second; third rostral segment shorter than first; the shape of humeral angles rounded; the shape of scutellar process elongated sub-cylindrically and apically rounded; legs dark, slender, and cylindrical; spongy fossula present in fore tibiae of males, absent in females; abdomen wide with urotergites exposed. Considering this information, we propose *T. yelapensis* sp. nov. as a member of the *T. phyllosoma* species group. Further phylogenetic analyses are needed to corroborate this hypothesis of the relationship.

### 3.6. Morphometry

Geometric morphometry. According to PCA and CVA analyses of factorial maps of the head shape for *T. yelapensis* sp. nov., *T. dimidiata s.s.*, *T. recurva*, and *T. gerstaeckeri*, *T. yelapensis* sp. nov constitutes an independent group that differs from other analyzed species. Total variation in the explained conformation was 98.1%. In the discriminant space, the first canonical variable represented 93.2% of the total conformation variation between groups (Figure 8). In this axis, *T. yelapensis* sp. nov. and *T. recurva* constitute fully isolated groups, whereas *T. gerstaeckeri* and *T. dimidiata* cluster together. The second canonical variable represented 4.9% of the total variation in the conformation, exhibiting low discrimination between analyzed species. Correspondence of each specimen to its own group was 92.8%, according to the K-means algorithm; however, 5/69 individuals of *T. dimidiata* and *T. gerstaeckeri* were discrepant with the initial plan of the group.

Linear morphometry. Univariate comparison of 14 morphological characters between *T. yelapensis* and *T. recurva* shows significant differences in 5 of them. These characters were length of anteocular region (ARL: *p* < 0.000), length of head excluding neck (HL: *p* < 0.000), length of first and second labial segments (Llb1: *p* < 0.03; Llb2: *p* < 0.001), and length of postocular region (PRL: *p* < 0.001) (Table 1).

### 3.7. Key to the Species of Triatoma recorded from Mexico (based on Lent and Wygodzisnky [7], Alejandre-Aguilar et al. [32], and Rengifo-Correa et al. [20])


1.Pronotum with humeral angles acute.

*T. mexicana*

-Pronotum with humeral angles rounded.
2
2.Large insects, longer than 25 mm; abdomen strongly widened, one-third as wide as body length
3
-Smaller insects, shorter than 25 mm, abdomen not strongly widened.
15
3.Head and thorax with abundant pilosity; first antennal segment attaining or surpassing the level of apex of clypeus; spongy fossulae absent in all tibiae of both sexes.
4
-Head and thorax appearing glabrous dorsally; first antennal segment rarely attaining but not projecting beyond the level of apex of clypeus; males with spongy fossulae present on fore or fore and mid tibiae; females lack spongy fossulae.
9
4.Pronotum black; connexivum largely dark brown, with small yellow or orange-red spot.
5
-Pronotum with posterior lobe extensively orange-yellow; connexivum largely orange-yellow, with a small anterolateral dark brown spot.

*T. picturata*

5.Corium largely black, with yellow or orange-red markings basally and subapical.
6
-Corium largely yellowish white, narrowly orange at base and black at apex

*T. pallidipennis*

6.Corium with delicate, suberect hairs about 0.5 mm long.
7
-Corium with slightly decumbent or adpressed setae not more than 0.3 mm long

*T. longipennis*

7.Hemelytra short, not extending beyond urotergite VI……………………………………….
8
-Hemelytra long, extending or almost extending to apex of abdomen.

*T. mazzottii*

8.Anteocular region 2.5X to 2.8X bigger than postocular; discal tubercle slighly elevated, lateral tubercle small; anterolateral angle slightly elevated.

*T. phyllosoma*

-Anteocular region 3.5X bigger than postocular; discal tubercle elevated, lateral tubercle very developed and pointed; anterolateral angle elongated and pointed.

*T. bassolsae*

9.Corium entirely piceous; connexivum along outer margin with narrow percurrent orange-red band, wider ventrally than on dorsum.

*T. recurva*

-Corium with light markings; connexivum with different color pattern.
10
10.Connexivum uniformly dark brown, or each segment with very small light-colored spot at junction of intersegmental suture of external margin; corium yellow at base and with wide transversal marking subapically.

*T. hegneri*

-Connexivum with large light-colored marks; corium marked differently.
11
11.Dark markings of connexival segments larger than orange-yellow markings.
*T. gerstaeckeri* (part)
-Dark markings of connexival segments smaller than orange-yellow markings.
12
12.Clavus completely black; venter uniformly black, except yellow or orange-yellow spot continuous from the external margin of connexivum to connexival suture, wider on the posterior portion of each segment and occupying almost 70% of each segment; corium glabrous, yellow with one central and basal black spot.
*T. yelapensis* sp. nov
-Clavus black basally, yellow to orange, yellow apically; venter uniformly light yellow, except dark pygophore, if extensively dark brown to black, then coloration of margin of venter adjacent to connexival suture yellow to orange yellow.
13
13.Venter largely light yellow, except dark pygophore; connections between each segment of the labium with a light yellow annulus, conspicuously contrasting with surrounding tegument.

*T. huehuetenanguensis*

-Venter largely piceous or dark brown, except yellow to orange yellow area adjacent to connexival suture; connections between each segment of the labium concolorous with labium, or at most slightly lighter than surrounding tegument.
14
14.Pronotum with anterior lobe with 1+1 discal tubercles, and 1+1 round, smaller lateral tubercles; membrane of hemelytra almost as pale as corium; pygophore almost round, as long as wide; postocular portion of head with sides subparallel..……..

*T. mopan*

-Pronotum with anterior lobe without distinctive tubercles; membrane of hemelytra distinctly darker than corium, rarely almost as pale as corium; pygophore ovoid, longer than wide; postocular portion of head rounded laterally.

*T. dimidiata*

15.Pronotum with red or yellowish markings in form of narrow band on collar on anterolateral angles and sides of pronotum, and on humeri and some portions along hind border of pronotum.

*T. sanguisuga*

-Pronotum without red or yellowish markings or humeral area lighter.
16
16.Pronotum with light markings (or faint) in humeral area.
17
-Pronotum without light markings.
19
17.Antennae with first segment elongate, attaining or slightly surpassing level of apex of clypeus; pronotum reddish brown or black, with lateral margins and humeral angles pale, very rarely entirely dark.

*T. rubida*

-Antennae with first segment short, falling distinctly short of apex of clypeus; pronotum different, in most cases unicolorous.
18
18.Pronotum with light markings on fore lobe and on humeral angles; head with orange-yellow markings of varied extension; anteocular region two and one-half times as long as postocular.

*T. nitida*

-Pronotum entirely dark except 1+1 light-colored spots on humeri; head dark brown to blackish; anteocular region 2 to 3 times as long as postocular.

*T. neotomae*

19.Synthlipsis two-thirds narrower than eye.

*T. brailovskyi*

-Synthlipsis wider than eye.
20
20.Neck uniformly dark.
21
-Neck with 1+1 lateral reddish or yellow spots.
25
21.Eyes in lateral view, surpassing the level of the lower surface and attaining the upper surface.
22
-Eyes very small, not attaining or not surpassing the level of the lower surface of the head.
23
22.Pronotum rugose and hairy; hemelytra with clavus black in the basal half and yellowish brown in the distal half. Corium dark brown, almost black, on most of the surface, with large yellowish spots at both basal and distal extremes. Membrane brownish gray with somewhat darker veins; legs shortly hairy.

*T. gomeznunezi*

-Pronotum rugose but not hairy; hemelytra with clavus black in the basal half and yellow in the distal half, with anterorlateral angle pointed. Corium with irregular dark spots brown. Membrane with very thin veins; legs without hairs.

*T. bolivari*

23.Head with slight arcuate depression dorsally behind clypeus; head elongated in lateral view, with eyes close to level of under surface of head; length of insects 13–23 mm.

*T. protracta*

-Head without arcuate depression behind clypeus, and relatively much shorter in lateral view, with eyes remote from level of under surface of head; length 9.5–13 mm.
24
24.General color black.

*T. peninsularis*

-General color brown, polished.

*T. sinaloensis*

25.Anterior lobe of pronotum with discal tubercles; anterolateral angles of pronotum prominent, stout, conical, or blunt; scutellum with elongate apical process.
26
-Anterior lobe of pronotum without discal tubercles; anterolateral angles not prominent rounded apically; posterior scutellar process short.
28
26.Connexivum with transversal yellow markings large closely approaching posterior border of segment and adjoining connexival suture

*T. infestans*

-Connexivum with narrow reddish markings along intersegmental sutures.
27
27.Size 23 mm or more; postocular portion of head with sides subparallel; pronotum with discal and lateral tubercles; fore femora relatively slender, from eight to nine times as long as it is wide; base of hemelytra conspicuously light-colored.

*T. gerstaeckeri*

-Size 22 mm or less; postocular portion of head distinctly rounded laterally; pronotum without lateral tubercles; fore femora relatively stout, about six times as long as wide; base of hemelytra only slightly lighter than remainder.

*T. indictiva*

28.Base of clypeus strongly swollen, conspicuously convex in lateral view; under surface of head sinuate in side view; venter evenly rounded.

*T. incrassata*

-Base of clypeus less swollen, flat above, its upper surface only very slightly convex in lateral view; under surface of head almost straight in lateral view; venter slightly flattened longitudinally along middle.

*T. barberi*


## 4. Discussion and Conclusions

Thirty-four species of Triatominae have been recorded in Mexico, of which nineteen consistently invade human houses; the remaining species are found only occasionally in association with humans, preferring sylvatic haenbitats [18]. *Triatoma yelapensis* sp. nov. increases the number of triatomines recorded for Mexico to 35. This species has shown potential as a *T. cruzi* vector because it was naturally infected with this parasite and it was collected regularly, from 2006 to 2015, in the peridomicile of a human settlement in Jalisco, Mexico. *Triatoma yelapensis* sp. nov. seems closely related to species of the *T. phyllosoma* species group. Hybridization is common between species of this species group [19], but *T. yelapensis* sp. nov. does not resemble any hybrid or valid species already described. *Triatoma yelapensis* sp. nov. differs from other species of *Triatoma* in external morphology—such as head morphology, corium and connexivum coloration pattern—and male genitalia.

*Triatoma yelapensis* sp. nov. shares several morphological characters with the *T. phyllosoma* species group members, particularly *T. recurva*. For *T. recurva*, Lent and Wygodzinsky [7] observed specimens “with orange-yellow outer margin dorsal and ventrally wider toward posterior portion of each segment; light color in some cases slightly extended mesad along intersegmental connexival sutures”. These characters are slightly similar to those exhibited by *T. yelapensis* sp. nov. However, the light portion of the connexivum of *T. yelapensis* sp. nov. is wider than it is for *T. recurva*, particularly toward the posterior portion of each segment, and is in a sub-triangular shape. The corium in *T. yelapensis* sp.nov. is yellow with a large dark central spot, whereas the corium of *T. recurva* is uniformly dark brown.

Head morphometry is also useful to characterize *T. yelapensis* sp. nov. conformation variables; the geometric morphometry of the head of triatomines is frequently used to provide statistical support for differences observed between similar species [33]. Here, we were able to detect differences between conformation variables of the head of *T. yelapensis* sp. nov., *T. dimidiata s.s.*, *T. gerstaeckeri*, and *T. recurva*. Additionally, we also found differences between the linear morphometry of the head of *T. yelapensis* sp. nov. and *T. recurva*. These differences were found in the length of the first and second labial segments, the length of anteocular and postocular regions, and the length of the head, excluding the neck. Linear morphometry of the head permitted us to see differences between very close species in the *T. phyllosoma* species group, such as those between *T. mopan* vs. *T. dimidiata* [4].

Lent and Wygodzinsky [7] and Zhao et al. [13] suggested that endosomal sclerites are potentially useful for species-level diagnosis. For instance, male genitalia was important in distinguishing *T. bassolsae* in Alejandre-Asguilar et al.’s 1999 study of *T. phyllosoma* (Alejandre-Aguilar et al., 1999). The size and shape of the endosomal process and the position of the denticles allow *T. yelapensis* sp. nov. to be distinguished from other species of the *T. phyllosoma* species group. The endosomal process of *T. yelapensis* sp. nov. is a medium size (0.45 mm) and discoid-shaped, with denticles in the anterior half (Figure 5 and Figure 6), whereas the endosomal processes of *T. gerstaeckeri* [34] and *T. recurva* are bigger than the endosomal process of *T. yelapensis* sp. nov., and denticles are numerous on all external margin (Figure 5); *Triatoma dimidiata* s.l has a small endosomal process (0.33–0.37 mm) and denticles are scarce or absent [35]. Phallosoma support is similar in most species analyzed, i.e., separated at the apex, except for *T. dimidiata s.s.*, which may also share similarities. The dorsal phallotecal sclerite is similar between all species, with a general oval shape with small variations, mainly in size [34,35].

## Figures and Tables

**Figure 1 insects-14-00331-f001:**
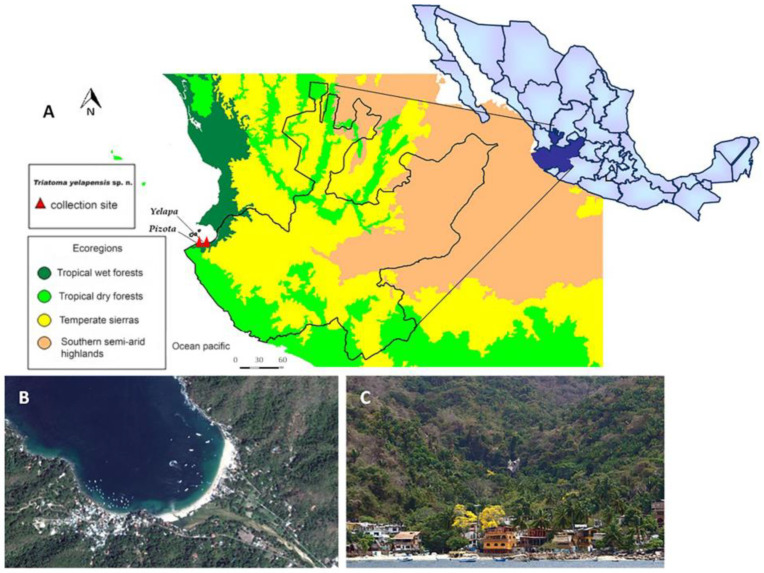
Geographic location of Yelapa and Pizota localities, municipality of Cabo Corrientes, Jalisco. (**A**) Location and ecoregions related to the collection points of *Triatoma yelapensis* n. sp. (**B**) Aerial view. (**C**) Panoramic view.

**Figure 2 insects-14-00331-f002:**
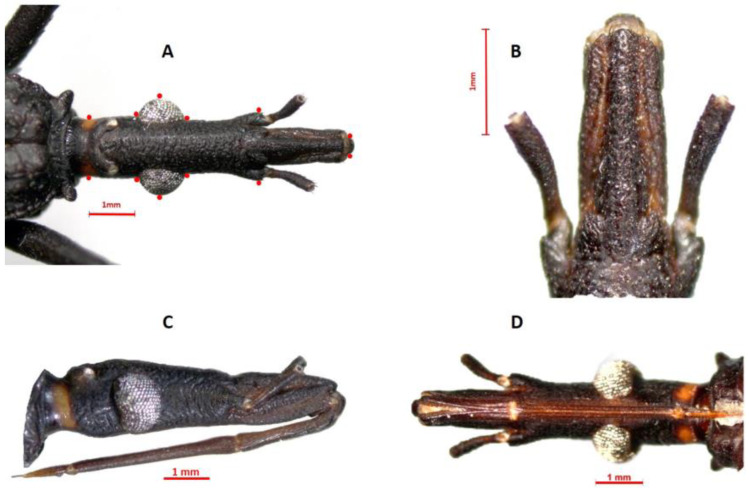
Head of *Triatoma yelapensis* sp. nov. (**A**) Dorsal view of the head showing the landmarks used for geometric morphometrics analysis, (**B**) details of the apex of the head in dorsal view, (**C**) head in lateral view, and (**D**) head in ventral view.

**Figure 3 insects-14-00331-f003:**
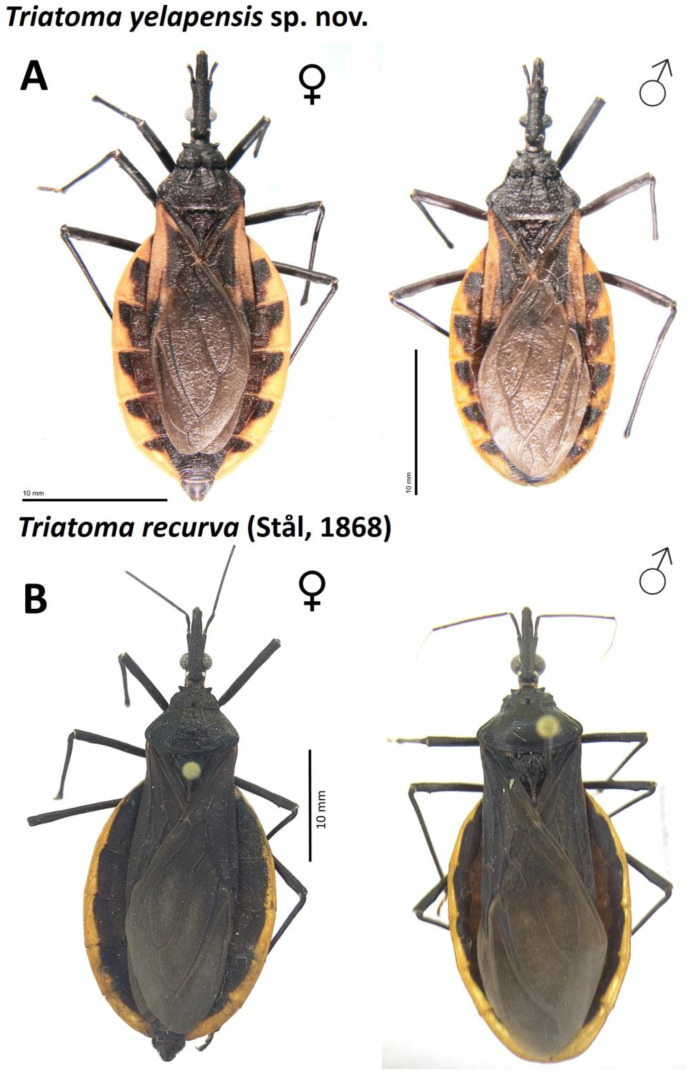
(**A**) *Triatoma yelapensis* sp. nov., female (paratype) and male (holotype) in dorsal view. (**B**) *Triatoma recurva*, female and male in dorsal view.

**Figure 4 insects-14-00331-f004:**
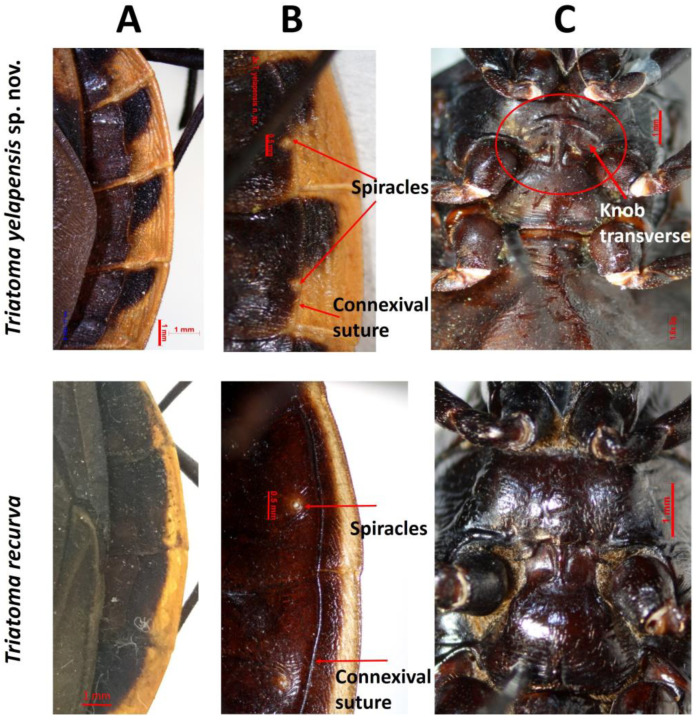
Connexivum and mesosternum of *Triatoma yelapensis* sp. nov. and *Triatoma recurva*. (**A**) Dorsal view of connexivum. (**B**) Ventral view of connexivum. (**C**) Ventral view of mesosternum.

**Figure 5 insects-14-00331-f005:**
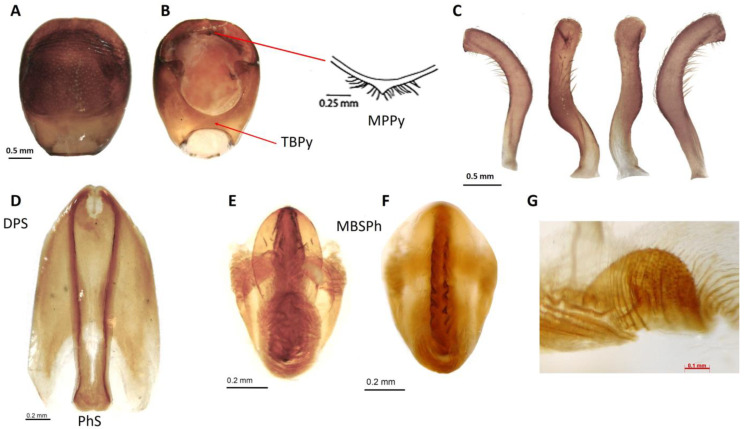
Male genitalia of *T. yelapensis* sp.nov. (**A**) Pygophore (ventral view); (**B**) pygophore (dorsal view); (**C**) parameres in ventral, inner, outer, dorsal views; (**D**) sclerites of phallosoma (ventral view); (**E**,**F**) sclerites of phallosoma (dorsal view); (**G**) endosomal process. Abbreviations: dorsal phallotecal sclerite (DPS); medial basal sclerite of phallosoma (MBSPh); median process of pygophore (MPPy); phallosoma support (PhS); transverse bridge of pygophore (TBPy).

**Figure 6 insects-14-00331-f006:**
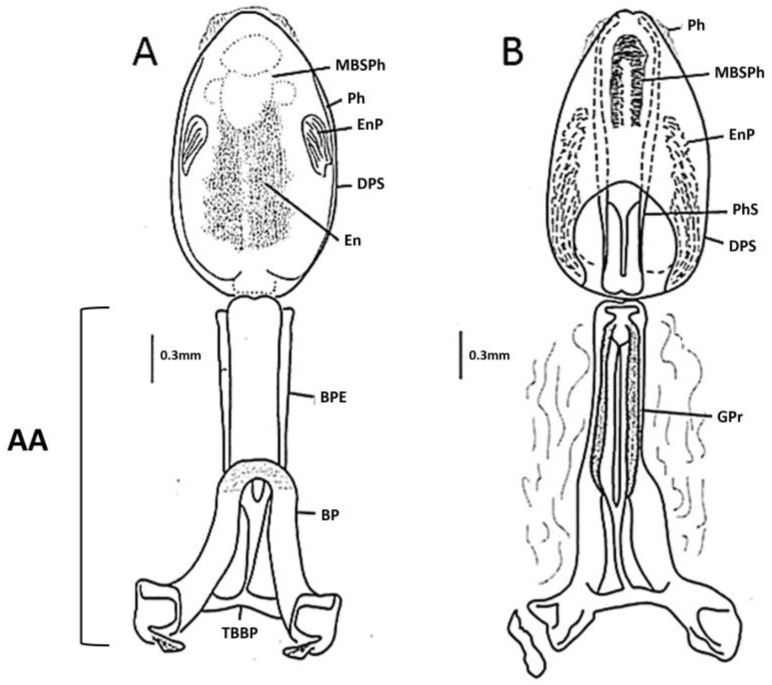
Phallus of *Triatoma yelapensis* sp. nov. (**A**) Dorsal view. (**B**) Ventral view. Abbreviations: articulatory apparatus (AA); basal plate (BP); basal plate extension (BPE); dorsal phallotecal sclerite (DPS); endosoma (En); endosomal process (EnP); gonopore process (GPr); medial basal sclerite of phallosoma (MBSPh); phallosoma (Ph); transverse bridge of basal plate (TBBP); phallosoma support (PhS).

**Figure 7 insects-14-00331-f007:**
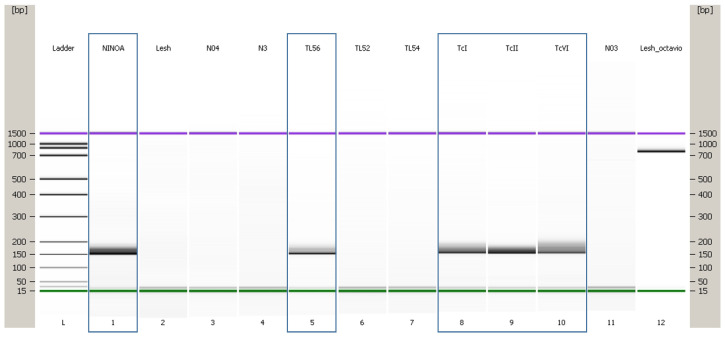
L = Ladder; 1. positive control NINOA (local); 5. TL56 (sample CAIMTriTp0009-*T. yelapensis*, positive); 8, 9, 10. TcI, II, and IV (cortesy of Dr. Bianca Zingales, Sao Paulo University, Brazil). Experiment performed in Agilent 2100 Bioanalyzer system^®^.

**Figure 8 insects-14-00331-f008:**
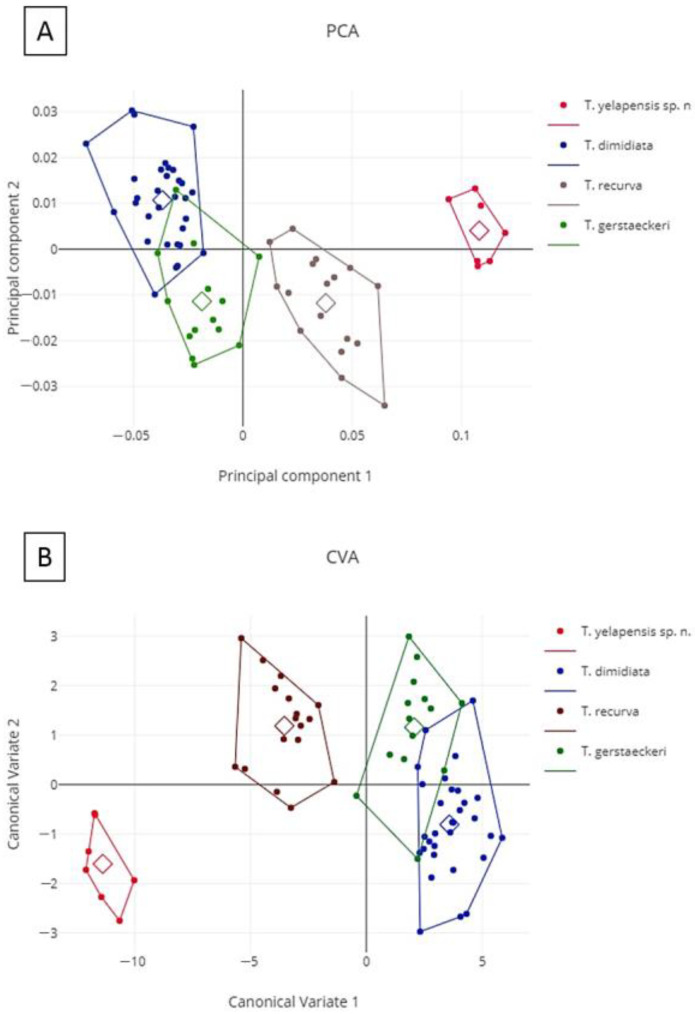
(**A**) Principal component (PCA) and (**B**) canonical variate (CVA) analyses of the factorial maps of the head shape for male and female specimens of *Triatoma yelapensis* n. sp., *T. dimidiata, T. gerstaeckeri*, and *T. recurva*.

**Table 1 insects-14-00331-t001:** *Triatoma yelapensis* character measures for females, males, the holotype and *T. recurva* (mm). ARL—length of anteocular region; bfla—length of basiflagellomere (the fourth was missing in all specimens); HL—length of head (excluding neck); lb1–lb3—length of first to third labial segments; Max = maximum value; Min = minimum value; PedL—length of pedicellus; PL—length of pronotum; PRL—length of postocular region; WE—width of eye; WH—width of head; TL—total length of the body (mm); SD = standard deviation; Sy—length of synthlipsis; WPPL—width of posterior pronotal lobe.

Character	*T. yelapensis*									*T. recurva*							
	Female (4 Specimens)				Male (3 Specimens)				Holotype CAIMTriTp-0006	Female (8 Specimens)				Male (9 Specimens)			
	Mean	SD	Min	Max	Mean	SD	Min	Max		Mean	SD	Min	Max	Mean	SD	Min	Max
TL	29.88	(1.31)	28	31	28.17	(1.04)	27	29	29	30.33	(1.28)	29	32	28.86	(0.69)	28	30
HL	5.63	(0.13)	5.45	5.76	5.55	(0.14)	5.40	5.67	5.67	5.01	(0.24)	4.65	5.39	4.81	(0.13)	4.61	4.99
WH	2.16	(0.07)	2.05	2.21	2.11	(0.14)	2.02	2.27	2.26	2.3	(0.15)	2.15	2.57	2.22	(0.17)	2.00	2.51
Sy	1.02	(0.05)	0.95	1.09	0.95	(0.04)	0.90	0.99	0.99	0.99	(0.04)	0.91	1.05	0.91	(0.06)	0.83	0.99
WE	0.58	(0.02)	0.56	0.60	0.60	(0.03)	0.56	0.62	0.62	0.66	(0.07)	0.57	0.82	0.67	(0.07)	0.56	0.78
ARL	3.51	(0.10)	3.36	3.58	3.44	(0.06)	3.37	3.48	3.48	2.96	(0.20)	2.63	3.26	2.77	(0.11)	2.62	3.01
PRL	1.09	(0.03)	1.06	1.13	1.05	(0.09)	0.98	1.16	1.16	0.95	(0.09)	0.86	1.12	0.95	(0.06)	0.86	1.03
WPP	5.84	(0.13)	5.71	5.99	5.97	(0.14)	5.82	6.10	6.09	6.25	(0.44)	5.60	6.84	6.03	(0.50)	5.40	6.74
PL	4.20	(0.24)	3.84	4.35	4.27	(0.09)	4.17	4.35	4.34	4.57	(0.35)	4.09	5.08	4.35	(0.37)	3.73	4.92
lb1	2.22	(0.16)	2.07	2.39	2.07	(0.33)	1.70	2.31	1.96	2.00	(0.14)	1.81	2.21	1.95	(0.15)	1.66	2.12
lb2	3.64	(0.05)	3.58	3.68	3.51	(0.10)	3.40	3.59	3.39	3.09	(0.15)	2.87	3.26	3.04	(0.17)	2.82	3.36
lb3	0.95	(0.04)	0.91	1.01	0.87	(0.07)	0.79	0.92	0.89	0.91	(0.05)	0.83	0.97	0.88	(0.05)	0.78	0.94
PedL	1.50	(0.06)	1.44	1.55	1.44	(0.03)	1.41	1.46	1.46	1.18				1.26	(0.08)	1.2	1.35
bfla	3.1				3.34	(0.16)	3.23	3.46	3.46	3.6							

## Data Availability

The authors confirm that all data are available in this paper.

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
