# Peer review of "Triatoma yelapensis sp. nov. (Hemiptera: Reduviidae) from Mexico, with a Key of Triatoma Species Recorded in Mexico"

_insects, 2023, doi:10.3390/insects14040331_

Round 1

Reviewer 1 Report

This paper describes one new species of the genus Triatoma (Hemiptera, Reduviidae) that is vectors of Chagas disease, showing diagnostic characters from closely related species with a key. This paper is of high-quality and an excellent taxonomic description of a new species.

Line 474. “according K-means algorithm” may be “according to K-means algorithm”. 

Author Response

Dear Reviewer,

We appreciate your comments about our manuscript. We accepted the change suggested. We change the Line 474 as suggested.

We hope our work fulfil your expectations.

Thank you so much,

The authors

Reviewer 1 – “This paper describes one new species of the genus Triatoma (Hemiptera, Reduviidae) that is vectors of Chagas disease, showing diagnostic characters from closely related species with a key. This paper is of high-quality and an excellent taxonomic description of a new species.”

Line 474. “according K-means algorithm” may be “according to K-means algorithm”. – Done

Reviewer 2 Report

Dear Editor and Authors

The manuscript entitled "Triatoma yelapensis sp. nov. (Hemiptera: Reduviidae) from Mexico, with a key to the Triatoma species recorded in Mexico" is a valuable contribution as it shows solid evidence of a new species of Triatoma, the key presented is efficient and very well elaborated. Just note some ambiguous concepts in the description that could be improved by comparing the size with other structures and in other cases by better describing the structure. 

Best regards

Author Response

Dear Reviewer,

We appreciate your constructive and positive comments about our manuscript. We accepted all changes, and improved some concepts as suggested.

We hope our work fulfil your expectations.

Thank you so much,

The authors

Reviewer 3 Report

Dear authors, I made suggestions directly in the text. I hope they help.

Author Response

Dear Reviewer

We appreciate your constructive and positive comments about our manuscript. We accepted all changes suggested. We improved Introduction, Methods, and Results sections considering your suggestions. We update information and add recommended references in introduction, we added details about landmarks, and we improve details of species description and figures in results. All changes are detailed in the following chart. Finally, we check English spelling in detail.

We hope our work fulfills your expectations.

Thank you so much,

The authors

Round 2

Reviewer 3 Report

.